# Influence of Intermolecular Interactions on Crystallite Size in Crystalline Solid Dispersions

**DOI:** 10.3390/pharmaceutics15102493

**Published:** 2023-10-19

**Authors:** Hua Huang, Yong Zhang, Yao Liu, Yufei Guo, Chunhui Hu

**Affiliations:** 1Medical College, Qinghai University, Xining 810001, China; huanghua19980426@163.com (H.H.); zhangyong20100507@163.com (Y.Z.); 13158395346@163.com (Y.L.); guoyufei1349@163.com (Y.G.); 2State Key Laboratory of Plateau Ecology and Agriculture, Qinghai University, Xining 810001, China

**Keywords:** crystalline solid dispersion, interactions, crystallite size, solubility

## Abstract

Crystalline solid dispersions (CSDs) represent a thermodynamically stable system capable of effectively reducing the crystallite size of drugs, thereby enhancing their solubility and bioavailability. This study uses flavonoid drugs with the same core structures but varying numbers of hydroxyl groups as model drugs and poloxamer 188 as a carrier to explore the intrinsic relationships between drug–polymer interactions, crystallite size, and in vitro dissolution behavior in CSDs. Initially, we investigate the interactions between flavonoid drugs and P188 by calculating Hansen solubility parameters, determination of Flory–Huggins interaction parameters, and other methods. Subsequently, we explore the crystallization kinetics of flavonoid drugs and P188 in CSD systems using polarized optical microscopy and powder X-ray diffraction. We monitor the domain size and crystallite size of flavonoids in CSDs through powder X-ray diffraction and a laser-particle-size analyzer. Finally, we validate the relationship between crystallite size and in vitro dissolution behavior through powder dissolution. The results demonstrate that, as the number of hydroxyl groups increases, the interactions between drugs and polymers become stronger, making drug crystallization in the CSD system less likely. Consequently, reductions in crystalline domain size and crystallite size become more pronounced, leading to a more significant enhancement in drug dissolution.

## 1. Introduction

The Biopharmaceutical Classification System (BCS) concept was first introduced by LENNE et al. in 1995 [1]. BCS Class II and IV drugs constitute up to 90% of drug candidates currently in development, accounting for approximately 60% of poorly soluble drugs on the global market [2,3]. Furthermore, the trend of poorly soluble drugs in marketed products is not expected to be alleviated significantly in the near future. The low water solubility of poorly soluble drugs adds complexity to their further development [4]. Consequently, enhancing the solubility and dissolution rate of active pharmaceutical ingredients (APIs) stands as one of the most significant challenges in drug development. It is becoming increasingly prevalent in new drug candidates [5,6]. It has evolved into one of the most active research areas in pharmaceutical formulation [7].

Solid dispersion (SD) is widely recognized as one of the most effective strategies for improving the solubility and bioavailability of poorly soluble drugs [8]. Depending on the crystallinity of the drug within the polymer matrix, SDs can be categorized as crystalline solid dispersions (CSDs) or amorphous solid dispersions (ASDs) [9]. In the modern pharmaceutical industry, by reducing the crystallite size of drugs through the preparation process, there is a significant change in the physical properties of the drug, thereby increasing drug solubility and bioavailability [10]. Galli et al. also used theories to establish a correlation between the reduction in the crystallite size of drugs and surface area, solubility, and dissolution rate [11,12,13]. As a thermodynamically stable system, the CSD system effectively reduces the crystallite size of drugs [14,15,16], thereby enhancing drug solubility and bioavailability [17]. For some drugs that crystallize rapidly, CSDs have a significant advantage over ASD due to their high crystallization tendency and superior molecular mobility [18,19]. According to the Noyes–Whitney dissolution equation, particle surface area and crystallite size are critical factors influencing drug dissolution rate and solubility [20,21].

It is well established that factors like steric hindrance [22], effective glass transition temperature [23,24], and intermolecular interactions [25] all influence drug crystallite size. However, while the first two have been studied more systematically, there needs to be more exploration of the intrinsic relationship between intermolecular interactions, crystallite size, and in vitro dissolution behavior in CSDs. Chen Zhen et al. concluded that poloxamer-based CSDs are able to simultaneously regulate crystallite size, crystal morphology, and crystallization rate, which is a remarkably promising capability in the improvement of dissolution of oral formulations [26]. Intermolecular interactions are believed to affect drug solubility by altering drug crystallite size [27,28]. The interactions between drugs and polymers mainly occur through hydrogen bonding [29], coupling [30], and acid–base interactions [31], with hydrogen bonding being the most common [32]. When drugs are dispersed in polymers, interactions such as hydrogen bonding between pharmaceuticals and polymers influence molecular mobility, thereby altering drug crystallization rates [33].

Flavonoid compounds are a significant class of biologically active substances in traditional Chinese herbs. They are present in various natural plants and exhibit pharmacological effects such as blood pressure reduction, lipid-lowering, and antioxidation [34]. They hold great promise in the field of medicine. However, most flavonoids fall into BCS Class II, characterized by poor water solubility and low bioavailability, which presents challenges in their development and formulation [35,36,37].

Preliminary studies have shown that flavonoids interact with polymers mainly by hydrogen bonding [38,39]. In this study, we selected flavone (PHE) as the core structure. We employed flavonoid drugs with varying numbers of hydroxyl groups, namely, flavone (PHE, Figure 1A), 3-hydroxyflavone (FLV, Figure 1B), chrysin (CHY, Figure 1C), baicalein (BAC, Figure 1D), luteolin (LUT, Figure 1E), quercitrin (QUR, Figure 1F), and myricetin (MYR, Figure 1G). We prepared them as CSDs with the triblock copolymer poloxamer 188 (P188, Figure 1H) via co-spray drying. Our investigation explored the intrinsic relationships among the flavonoid drugs with varying hydroxyl group numbers, the intermolecular interactions with P188, crystallite size, and in vitro dissolution behavior.

## 2. Materials and Methods

### 2.1. Materials

Flavone (PHE, purity > 99%), 3-Hydroxyflavonol (FLV, purity > 95%), Chrysin (CHY, purity > 98%), Baicalein (BAC, purity > 98%), Luteolin (LUT, purity > 98%), Quercetin (QUR, purity > 98%), and Myricetin (MYR, purity > 98%) were acquired from Beijing Coupling Technology Co. Ltd. (Shanghai, China). Poloxamer 188 (P188, purity > 99%) was acquired from Shanghai, China. Methanol (purity > 99.5%), anhydrous ethanol (purity > 99.7%), and acetone (purity > 99%) were procured from Tianjin Daimao Chemical Reagent Technology Co. Ltd. (Tianjin, China). Chromatography-grade methanol (chromatographic grade) was acquired from Merck Co., Ltd. (Darmstadt, Germany). Chemical structures and pertinent physical and chemical characteristics of the drugs and polymers employed in our experiments are highlighted in Figure 1.

### 2.2. Instruments and Methods

#### 2.2.1. Generation of Flavonoid Drugs and P188 Samples

Various combinations of flavonoid drugs with P188 (10/90, 30/70, 50/50, 70/30, 90/10, *w*/*w*) were dissolved in dichloromethane to achieve a total concentration of 5 wt%. The resulting solution was then introduced into a Yamato spray dryer (ADL 311S, Yamato Scientific Company, Ltd., Tokyo, Japan). This spray dryer operated with an inlet temperature of 55 °C, an outlet temperature of 32 °C, a solution feed rate of 4 mL/minute, and an atomizing N_2_ pressure of 0.1 MPa. After the spray-drying process, the flavonoid drugs-P188-CSDs were subjected to vacuum drying for a minimum of 24 h and stored in a room-temperature desiccator.

The mixtures of flavonoid drugs and P188 were generated by sieving the two sample powders to guarantee mixing uniformity. 

#### 2.2.2. Theoretical Calculation of Drug–Polymer Solubility Parameters

Interaction parameters were determined using Cohesive Energy Density (Equation (1)) [40]. The applicability of Equation (1) is limited to nonpolar solute–solvent interactions; to find a system that is suitable for polar or where hydrogen-bonding interactions are present, the Hansen solubility parameterization method is proposed [41]. The Hansen solubility method is a widely employed technique for assessing the miscibility of drug–polymer combinations. It relies on computed solubility parameters (Δδ_p_) to gauge the compatibility between molecules based on their dissimilarity [42]. To compute the overall solubility parameters of both the drugs and polymers, Equation (2) was utilized [41]. The partial solubility parameters of a particular substance were calculated using the group contribution method (Equation (3)).
(1)$$\delta ={\left(\mathrm{CED}\right)}^{0.5}={(∆E_{V}/V_{m})}^{0.5}$$
(2)$$\delta_{t}=\delta_{d}^{2}+\delta_{p}^{2}+\delta_{h}^{2}$$
(3)$$\delta_{d}=\frac{\sum F_{d}}{V};\;\delta_{p}=\frac{\sqrt{\sum F_{p}^{2}}}{V};\delta_{h}=\sqrt{\frac{\sum E_{h}}{V}}$$
where ΔE_v_ denotes the evaporation energy; V_m_ signifies the molar volume of the substance; δ_p_, δ_h_, and δ_d_ denote the solubility parameters associated with dispersion, polarity, and hydrogen bonding, respectively; F_d_ stands for the molar absorption constant pertaining to the dispersion group; F_p_ corresponds to the molar absorption constant for the polar group; E_h_ represents the hydrogen-bonding energy; and V represents the molar volume of the substance.

#### 2.2.3. Differential Scanning Calorimetry (DSC)

An annealing method initially devised by Yu was employed to identify the solubility of the crystalline drugs within polymers [43]. Specifically, drug–polymer mixtures were allowed to anneal at diverse temperatures to generate phase equilibrium before scanning via DSC (STA49F3-DSC200F3, Netzsch, Germany) to detect residual drug crystals. This method provided both the upper and lower limits for the equilibrium solution temperature for a drug–polymer mixture annealed at various temperatures. The Flory–Huggins model was utilized to determine the interaction parameters between the drug and polymer [14]. The drug’s activity at a particular solubility level was determined using Equation (4), while the drug–polymer interaction parameter was calculated using the Flory–Huggins model as described in Equation (5). A DSC approach was employed to ascertain the following parameters.
(4)$$\ln\alpha_{1}=\frac{∆H_{m}}{R}(\frac{1}{T_{m}}-\frac{1}{T_{m0}})$$
(5)$$\ln\alpha_{1}=\ln\Phi_{1}+\left(1-\frac{1}{x}\right)\Phi_{2}+\chi \Phi_{2}^{2}$$
where α_1_ stands for drug activity, T_m0_ is the melting temperature of the pure drug, ΔH_m_ represents the molar heat of fusion of the pure drug, T_m0_ represents the solubility temperature, Φ_1_ denotes the volume fraction of the drug, Φ_2_ denotes the volume fraction of the polymer, x represents the molar volume ratio of the polymer to the drug, and χ signifies the drug–polymer interaction parameter. 

#### 2.2.4. Nuclear Magnetic Resonance (^1^H-NMR)

To investigate the molecular mechanism of the drug–polymer interaction, the flavonoid drugs, P188, and the different drug-loaded SDs were dissolved in dimethyl sulfoxide-d6+TMS (0.03) (TMS is tetramethylsilane, and TMS is commonly used as an internal standard for chemical shifts), and their ^1^H-NMR spectra were recorded at room temperature using an AVANCE NEO 400 spectrometer (Bruker BioSpin GmbH, Rheinstetten, Germany). Dimethyl sulfoxide-d6+TMS (0.03) (DMSO-d6) was employed as a reference for solvent signals. Spectral assignments for flavonoid drugs and P188 were carried out based on previous studies. In the experimental design, the mass of the pure drug was consistently kept at 10 mg throughout the experiment in order to avoid multiple variables in the measurements, which would ultimately lead to errors caused by other factors.

#### 2.2.5. Fourier-Transform Infrared Spectroscopy (FT-IR)

To assess the involvement of different functional groups in the compounds involved in drug–polymer interactions, IR spectra of SDs were recorded using FT-IR spectroscopy (Nicolet 6700, Thermo, Waltham, MA, USA) with a spectral resolution of 4 cm^−1^ [44]. The wavenumbers of IR spectra ranging from 4000 to 400 cm^−1^ were recorded for subsequent analysis. Before testing, the samples were equilibrated at room temperature and subjected to vacuum drying for 24 h to minimize the impact of moisture on subsequent analysis. The FT-IR spectra observed for pure drugs corresponded with those reported in the literature [45,46,47,48,49].

#### 2.2.6. Polarized Optical Microscopy (POM)

POM (Leica DM4P, Heidelberg, Germany) was utilized to observe flavonoid drug, raw P188, and CSD morphology. The raw flavonoid drugs, P188, and CSDs were dissolved in methanol and deposited onto a glass substrate using spin-coating at 2000 rpm (KW-4A, Beijing, China) for a duration of 9 s. Subsequently, the crystallization of the spin-coated film was analyzed at room temperature.

#### 2.2.7. Powder X-ray Diffraction (PXRD)

Powder X-ray diffraction (PXRD) patterns were obtained using an ESCALAB™ XI+ X’pert Powder X-ray Diffractometer (ESCALAB™ XI+, Waltham, MA, USA). The samples were positioned within a zero-background silicon sample holder. PXRD experiments were conducted employing an automatic divergence slit graphite monochromator (with a 0.2 mm receiving slit) and continuous scanning in the 5° to 35° (2θ) range at a scanning rate of 16°/min, with a scanning step size of 0.02°/2θ. The system was operated at a voltage setting of 40 kV and a current of 200 mA for the CSD system, and the scanning was performed over a 5-day period to capture the kinetics of crystallization [15]. Following the complete crystallization of the drug after 20 days, the crystalline domain size (CDS) was computed using an identical scanning range, a scanning speed of 1°/minute, and a scanning step of 0.01°/2θ. The diffraction peak was measured using the Scherrer equation (Equation (6)), indicating that the CDS of the crystallite was perpendicular to the reflection [24]. We employed the Scherrer equation as follows:(6)$$\tau =K\;\lambda /(\beta_{\tau }\;\cos\theta )$$
where *τ* denotes the average crystallite dimension oriented perpendicular to the (hkl) reflection plane, K is a shape factor (usually around 0.9), λ stands for the X-ray wavelength (1.54 Å), and β*_τ_* represents the line broadening at half the maximum intensity. Additionally, θ represents the Bragg angle.

#### 2.2.8. Particle-Size Analyzer (PSA)

The particle size and size distribution of CSDs were assessed using a Mastersizer 2000 (Malvern Instruments, Malvern, UK) under dry test conditions. The method used was dry testing. Approximately 1.0 g of each sample was placed in the dry test sample cell, and a refractive index (RI) of 1.00 was applied for the dispersion medium. The RI of PHE, FLV, CHY, BAC, LUT, QUR, and MYR was 1.635, 1.679, 1.699, 1.732, 1.768, 1.823, and 1.864, respectively, and an absorption index of 0.01 was applied. The sample detection duration was set to 10 s. The dispersion pressure was maintained at 2.0 bar, with an injection speed of 50%, and a slit width of 1.5 mm, while the shading ranged from 0.001 to 6.0%.

#### 2.2.9. Powder Dissolution

The dissolution characteristics of the powder formulations were assessed under non-sink conditions using a ZRS-8G dissolution apparatus (Tianjin Jingtuo Instrument Technology Co., Ltd., Tianjin, China). Powder dissolution experiments were precisely weighed in terms of the samples to be tested with a mass of 45 mg of each of the pure flavonoids. The dissolution medium consisted of 900 mL of PBS (pH = 7.4), and the dissolution process extended over 4 h. Stirring was performed with a paddle at 75 rpm, maintaining a temperature of 37 °C. Sampling occurred at time intervals of 15, 30, 60, 120, and 240 min. After dissolution, 0.3 mL of the sample was taken, and a blank dissolution medium of the same temperature and volume was added, centrifuged at 13,000× *g* rpm for 3 min, and subjected to double dilution with methanol for drug assessment. Drug concentration was determined by using HPLC.

High-performance liquid chromatography (HPLC) (Agilent 1260 Series, USA) was employed for the quantification of flavonoid drugs. A Diamonsil C18 column (4.6 × 150 mm, 5 μm) was utilized with a mobile phase consisting of methanol/water (70/30, *v*/*v*) at a flow rate of 1.00 mL/minute. The column temperature was set at 30 °C, and the injection volume was 10.00 μL. UV-detection wavelengths for PHE, FLV, CHY, BAC, LUT, QUR, and MYR were 265, 350, 270, 280, 348, 374, and 375 nm, respectively. The HPLC method underwent validation for specificity, calibration curve, precision, repeatability, stability, and recovery.

#### 2.2.10. Statistical Analysis

Statistical analysis was conducted using SPSS 22.0 software. Measurement data were presented as mean ± standard deviation. Multiple samples were compared using ANOVA, with a significance level set at *p* < 0.01. Graphs and figures were created using GraphPad Prism 8.3.0 software. Pharmacokinetic parameters were assessed using a non-compartment model analysis in DAS 3.2.8.

## 3. Results and Discussion

### 3.1. Evaluation of Intermolecular Interactions between Drugs and P188

To assess compatibility, we utilized the difference in partial solubility parameters, Δδp, between drugs and P188. A smaller Δδp signifies closer alignment of partial solubility parameters between the drug and polymer, indicating better compatibility (∆δ < 7.0 MPa^1/2^) [50]. As shown in Figure 2A, flavonoid drugs exhibit good miscibility with P188. Moreover, as the number of hydroxyl groups increases, the compatibility gradually improves, with the order of compatibility being MYR-P188 (2.54) > QUR-P188 (2.62) > LUT-P188 (2.70) > BAC-P188 (2.78) > CHY-P188 (2.86) > FLV-P188 (2.94) > PHE-P188 (3.01 (J·cm^−3^)^1/2^).

Hansen solubility parameters are the most reliable method for evaluating drug–polymer compatibility, yet they overlook chain conformation and directional or long-range interactions like hydrogen bonding [51]. Hence, further verification is needed to enhance the predictive accuracy.

Based on Flory–Huggins solution theory, assessing drug–polymer interactions is crucial for evaluating system homogeneity. The χ values are constant thermodynamic parameters determined by the chemical structure of the drug and polymer. When χ < 0, it signifies that the drug–polymer interaction is stronger than the drug–drug interaction, indicating good miscibility within the drug–polymer system. Conversely, when χ > 0, the drug tends to form isolated aggregates within the system, leading to phase separation and reduced miscibility [52]. In our study, we determined equilibrium melting temperatures (*T_c, onset_*) and Flory–Huggins interaction parameters at various scanning rates after mixing flavonoid drugs with P188. As illustrated in Figure 2B, the interactions between flavonoid drugs and P188 (excluding FLV) are strengthened with increasing hydroxyl groups. The trend in interaction strength is as follows: MYR (−4.3128) > QUR (−3.9708) > LUT (−3.5416) > BAC (−2.7912) > CHY (−2.1431) > PHE (−1.1078) > FLV (−0.4876). The measured χ values align closely with the predicted values derived from the Hansen solubility parameter, indicating that increased hydroxyl groups lead to stronger interactions between flavonoid drugs and P188.

In the ^1^H-NMR study, if there are interactions between flavonoid drugs and the polymer, adding the polymer can broaden, shift, or eliminate specific hydroxyl peaks in the drug, revealing these interactions through changes in chemical shift [53]. We assessed these interactions by observing the changes in the liquid-state ^1^H-NMR chemical shifts of hydroxyl groups in flavonoid drugs at different drug loading levels in CSDs [54]. As depicted in Figure 3, the chemical shifts observed for pure drugs corresponded with those reported in the literature [48,55,56,57]. Analyzing the hydroxyl chemical shifts of drugs, peak shape or position variations occurred with changing drug loading in CSDs. Among these changes, Δδ represents the chemical shift change of drugs relative to CSDs (Equation (7)).
(7)$$∆\delta (\mathrm{ppm})=\delta_{\mathrm{CSDs}}-\delta_{\mathrm{drug}}$$
where Δδ stands for alterations in chemical shifts of the CSD system compared to the drug, δ_drug_ is the chemical shift of the drug, and δ_CSDs_ is the chemical shift of the CSD system with various drugs loaded. 

The displacements between PHE and P188 are so negligible that they can be disregarded, implying an absence of interactions between them (Figure 3A–C). In the case of FLV-P188, the peak of 3-OH broadens as drug loading changes, indicating the presence of stronger hydrogen-bonding interactions between the FLV’s hydroxyl group and P188 (Figure 3D–F). CHY, which has two hydroxyl groups, shows no shift of 5-OH, but a similar peak broadening occurs in 7-OH, resembling the FLV pattern. This implies the existence of hydrogen-bonding interactions between CHY and P188 (Figure 3G–I).

As the number of drug hydroxyl groups increases to three (Figure 3J–L), BAC exhibits two broad peaks of H around 9.74 ppm at low drug loading. With increased drug loading, these peaks gradually separate into two distinct peaks (6-OH and 7-OH). This suggests that at low drug loading, due to the presence of a significant amount of P188, the drug’s ability to form hydrogen bonds with P188 is strong. However, an increase in drug loading implies a reduction in P188 quantity and a weakening of the drug’s ability to form hydrogen bonds, resulting in two peaks, similar to other flavonoid drugs. Similar to BAC, the greater the number of hydroxyl groups, the more pronounced the change, indicating stronger hydrogen-bonding interactions between flavonoid drugs and P188 (Figure 3M–U). According to the experimental results of ^1^H-NMR, the chemical shifts of flavonoids prepared with P188 into the system of CSDs differ at different loading capacities. The results exhibited a more pronounced change in the chemical shifts of 7-OH, 6-OH, and 4′-OH in the presence of multiple hydroxyl groups concomitantly, with 7-OH appearing to be the primary binding site for the drug–P188 interactions, followed by the 4′-OH position. In the structure of flavonoids with three six-membered rings, the a-ring is a conjugated system (Figure 1A); the introduction of the hydroxyl group can form intramolecular hydrogen bonding, which increases the stability of the system; 7-OH is located in the 4-carbonyl position of the opposite position, susceptible to the conjugation of the carbonyl group induced by the p–π conjugation of the influence of the H^+^, which leads to easier separation [58]. The 5-OH is influenced by the 4-carbonyl group to form intramolecular hydrogen bonds, where the H^+^ of the phenolic hydroxyl group is not easily ionized out [59]. The stronger the hydrogen-bonding activity of a substance, the better its ability to act as a hydrogen-bond donor and form intermolecular hydrogen bonds with other hydrogen-bond-acceptor substances, which can interact with polymers to a greater extent [60], where 7-OH is the main binding site for drug–P188 interaction. It is noticeable that there was no significant change in the chemical shift of 5-OH in the CSD systems of all flavonoids [57]. Meanwhile, the a-ring is in the same plane as the b-ring, in which 3-OH can form an intramolecular hydrogen bond with the 4-carbonyl group [61]. The c-ring is attached to the b-ring by a single bond at C-2 and can be rotated [62]. When the hydroxyl group is present in the c-ring, it is affected by the electron-withdrawal effect of -OH. When the number of -OH gradually increases, the oxygen on the hydroxyl group has an unshared electron pair and can form a p–π conjugation with the benzene ring, which transfers the p-electrons on the oxygen to the benzene ring, resulting in easier ionization of H^+^, with 4′-OH > 3′-OH ≈ 5′-OH in the c-ring [57,63].

From the results of the ^1^H-NMR experiments, it is evident that interactions exist in the CSD systems of flavonoid drugs at different drug loading levels. Moreover, lower drug loading levels result in more noticeable chemical shift changes, indicating stronger interactions [64].

Previous studies have confirmed that there are some differences between the interactions in solids and their interactions in liquids. Considering that the data in the solid state are more convincing, we first used ^1^H-NMR to confirm the liquid interactions, and then, we mainly used the solid state for the data study of their experimental results. To further investigate the strength of the degree of crystallization between flavonoid drugs and P188, changes in the solid-state degree of crystallization were compared using FT-IR absorption peak shifts, intensities, and shapes [65,66]. Differences in the degree of crystallization are also partially due to differences in the interactions [26]. It was verified from the perspective that the difference in interactions affects their crystallization behavior. Both flavonoid drugs and P188 contain multiple -OH peaks, and to avoid interference, we selected the -C=O peak in flavonoid drugs as the focus of our study. Additionally, as each flavonoid drug possesses unique physicochemical properties, we intend to use the percentage change in the difference between the -C=O peak shift in flavonoid drugs relative to that in the CSDs (∆σ (%)) to elucidate the interactions (Equation (8)).
(8)$$∆\sigma \left(\%\right)=(\sigma_{\mathrm{CSDs}}-\sigma_{\mathrm{drug}})/\sigma_{\mathrm{drug}}$$
where ∆σ is the rate of wavelength change in the CSD system relative to the carbonyl group in the drug, σ_drug_ is the wavelength variation in carbonyl groups in the drugs, and σ_CSDs_ is the wavelength variation in carbonyl groups in the CSD system with 30% drug loading. 

Compared to pure flavonoid drugs, the CSD system with a drug loading of 30% exhibits more noticeable shifts in the -C=O peak position (indicated by black arrows in Figure 4). Calculating the percentage change in the shift ratio of the carbonyl peak (∆σ (%)) reveals a positive correlation between the number of hydroxyl groups and interactions within the CSD system, with the following magnitudes: MYR (0.5301 ± 0.0652%) > QUR (0.4876 ± 0.0811%) > LUT (0.1746 ± 0.0218%) > BAC (0.1420 ± 0.0160%) > CHY (0.1180 ± 0.0050%) > PHE (0.1120 ± 0.0002%) > FLV (0.0690 ± 0.0012%). This suggests that flavonoid drugs may form hydrogen-bonding interactions (-C=O⋯H-O) with P188.

The above results demonstrate the presence of interactions between flavonoid drugs and P188, with interactions increasing with an increasing number of hydroxyl groups (except for FLV). In the following subsections, we elucidate the relationship between interaction forces and crystallite size changes, focusing on crystallization kinetics, crystalline domain size, and crystallite size variations.

### 3.2. Investigation of Drug Crystallite Size in CSDs

#### 3.2.1. Crystalline Growth Rate

To explore the influence of interactions between P188 and flavonoid drugs with varying hydroxyl group numbers on crystalline growth rates, we examined the effect of drug loading (3% and 30%) on P188 and the impact of P188 on the crystalline growth rate of flavonoid drugs.

As shown in Figure 5A, pure P188 exhibits a rapid crystallization rate. With the addition of drugs, the growth rate of P188 decreases. Moreover, as the number of hydroxyl groups in drugs increases, the crystallization rate of P188 slows. The order of crystallization rates is as follows: P188 (46.30 ± 0.17) μm·s^−1^ > PHE (33.92 ± 0.55) μm·s^−1^ > FLV (31.28 ± 0.38) μm·s^−1^ > CHY (27.64 ± 0.27) μm·s^−1^ > BAC (27.07 ± 0.30) μm·s^−1^ > LUT (24.06 ± 0.34) μm·s^−1^ > QUR (23.30 ± 0.29) μm·s^−1^ > MYR (22.10 ± 0.35) μm·s^−1^. Each flavonoid drug exhibits a significant difference in crystallization growth rate compared to P188 (*p* < 0.01). A chi-square trend test was conducted to explore whether significant differences exist among the flavonoid drugs, revealing a significant enhancement in the inhibition of crystalline growth on P188 with increasing interactions (*p* < 0.01).

We assessed the crystallization rates of flavonoid drugs in CSD systems with a drug loading of 30% (Figure 5B). All pure flavonoid drugs exhibited rapid crystallization within 30 s, making it difficult to differentiate the rates, suggesting similar growth rates. Crystal growth within seven days in the CSD systems showed a linear trend. The growth rates of drug crystals decreased with increasing interactions, with the following trends: PHE (352.0 ± 0.59779) μm·min^−1^ > FLV (180.6 ± 0.59099) μm·min^−1^ > CHY (5.484 ± 0.19713) μm·min^−1^ > BAC (0.2296 ± 0.01421) μm·min^−1^ > LUT (0.1598 ± 0.00634) μm·min^−1^ > QUR (0.0034 ± 0.00005) μm·min^−1^ > MYR (no crystallization within seven days).

#### 3.2.2. Crystallization Kinetics and Crystalline domain Size

We utilized PXRD to study the crystallization kinetics and crystalline domain size changes of flavonoid drugs in CSDs. Firstly, it was evident by PXRD that no transcrystallization occurred in CSDs for several flavonoids, which ensured the consistency of the evaluation. Except for MYR-CSDs, crystalline nuclei appeared in the mixture for all other CSDs within five days. In the CSD system with a drug loading of 30%, notable differences in the crystallization rates of various flavonoid drugs were observed. Crystalline nuclei appeared in the mixture for FLV, PHE, CHY, BAC, LUT, and QUR at 0 min, 3 min, 3 min, 6 min, 3 h, and 8 h, respectively, and MYR showed no crystalline nuclei within five days, indicating that more extensive interactions result in slower drug crystallization. It must be explained that, in such systems, different dynamic processes compete, and the more different interactions are possible, the slower the equilibrium is achieved [67].

According to Scherrer’s equation (Equation (6)), the decrease in the size of crystalline domains can make the PXRD diffraction peaks broader. In the previous study of our group, the researchers also used Scherrer’s equation several times to confirm the relationship between the reduction in drug crystalline domain and crystallite size in CSDs [24]. After leaving the CSDs mentioned above at room temperature for 20 days until complete crystallization, we conducted a PXRD analysis and calculated changes in crystalline domain size. The results revealed a trend of decreasing crystalline domain size for all drug compounds within the CSDs, with a more significant decrease in crystalline domain size associated with increased interactions. The percentage change in crystalline domain size (∆*τ* (%)) (Equation (9)) was found to be directly proportional to the interactions (Figure 6U). The trends in crystalline domain size change for flavonoid drugs within the CSD systems were as follows: MYR (53.56 ± 15.76%) > QUR (43.56 ± 6.47%) > LUT (24.29 ± 10.75%) > BAC (21.18 ± 5.12%) > CHY (17.63 ± 6.77%) > PHE (3.271 ± 11.585%) > FLV (1.491 ± 0.47%). FLV exhibited anomalous results in many cases, possibly due to its rapid crystallization, which might hinder the formation of intermolecular interactions with P188. The crystallization kinetics results from Figure 6 can be corroborated with those of Figure 5.
(9)$$∆\tau \left(\%\right)=(\tau_{\mathrm{drug}}-\tau_{\mathrm{CSDs}})/\tau_{\mathrm{drug}}$$
where ∆*τ*(%) is the rate of change of crystalline domain size in the CSD system concerning the drug, ∆*τ*_drug_ is the crystalline domain size of the drug, and ∆*τ*_CSDS_: is the crystalline domain size of the CSD system with 30% drug loading.

In the experiment, we can observe that the addition of the drug delays the appearance of the polymer nuclei and also inhibits the growth of the nuclei [68]. Chen also confirmed that when acetaminophen polymer forms hydrogen bonds, there is an interaction between the drug and the polymer, which affects the nucleation and the growth of the drug, and the stronger the interaction, the more obvious this effect is [26].

#### 3.2.3. Particle-Size Analysis

The results obtained from laser-particle-size analysis for CSD systems with a drug loading of 30% align with the findings for crystalline domain size (Figure 7A). That is, all CSDs exhibited a reduction in crystallite size. The percentage change in crystallite size (∆Dv (%)) of flavonoid drugs within the CSD system with a drug loading of 30% compared to the crystallite size of pure drugs followed this trend: MYR (87.92%) > QUR (77.26%) > LUT (58.38%) > BAC (41.62%) > CHY (19.95%) > PHE (5.448%) > FLV (3.788%). As the interactions increased, the ∆Dv (%) in the CSD system gradually increased.

Because CSDs are a composite of drugs and P188, for a more accurate assessment of the actual crystallite size of flavonoid drugs, we utilized the difference in solubility between drugs and P188. We employed a water-washing method to remove P188, enabling a more direct measurement of the crystallite size of flavonoid drugs (Figure 7B). After water washing, CSDs, except for FLV, exhibited the same trend: MYR (90.24%) > QUR (89.89%) > LUT (81.12%) > BAC (47.80%) > CHY (36.88%) > FLV (30.19%) > PHE (27.35%). The reduction in crystallite size of flavonoid drugs relative to pure drugs within the CSD system increased consistently with increasing interactions, mirroring the trend observed in domain size changes.
(10)$$∆D_{v}\left(\%\right)=({D(0.5)}_{\mathrm{drug}}-{D(0.5)}_{\mathrm{CSDs}}{)/D(0.5)}_{\mathrm{drug}}$$
where ∆D_v_ (%) represents the CSD system relative to the rate of change in diameter equivalent to 50% of the cumulative distribution of drug particle size (0–100%), D(0.5)_drug_ is the diameter equal to 50% of the cumulative distribution of the particle size of the drug (0–100%), and D(0.5)_CSDs_ is the diameter equivalent to 50% of the cumulative distribution of the particle size of the drug loaded with 30% or washed with water (0–100%).

Based on the results obtained from POM, PXRD, and PSA, it is evident that as the interactions increase, the crystallization growth rate of the drug and P188 becomes slower, leading to larger percentage changes in crystalline domain size (∆*τ* (%)) or crystallite size (∆Dv (%)).

### 3.3. Powder Dissolution

The ultimate goal of this study is to enhance the drug dissolution rate by preparing CSD formulations. Therefore, we conducted powder dissolution experiments on the prepared CSDs (Figure 8). The research findings indicate that the cumulative percentage dissolution rates at 4 h for CSD systems with a drug loading of 30% are all higher than those for pure drugs. To provide a more intuitive comparison of the dissolution-enhancing effect of CSD systems on flavonoid drugs, we selected the percentage change in cumulative dissolution at 4 h, denoted as ∆S (%), as an index for comparison (Equation (11)). The results show the following trend: QUR (1653.92 ± 231.21%) > LUT (571.51 ± 28.62%) > MYR (388.12 ± 12.43%) > BAC (315.07 ± 12.02%) > CHY (211.76 ± 45.43%) > FLV (188.85 ± 6.92%) > PHE (141.19 ± 7.07%) (Figure 8H). ∆S (%) is directly proportional to the degree of interaction. The higher ∆S (%) for FLV with a drug loading of 30% compared to PHE may be due to rapid crystallization with P188 in the crystallization kinetics. Notably, MYR’s lower dissolution is inconsistent with the changing trend observed in other drugs. The lower degree of dissolution of MYR may be due to the following reasons: (1) Research suggests that its in vitro dissolution is related to drug loading [69]. While 30% of drug loading was selected in order to standardize the drug loading in this study, it may not be the optimal drug loading in the dissolution of MYR under this condition, which does not reflect the optimal dissolution behavior; (2) The powder dissolution behavior of a drug is affected by a number of factors, such as the particle size of the drug, the nature of the drug, with dissolution conditions, etc. [10]. Whereas this study mainly illustrates the variability in terms of variation in drug particle size, there may be other factors that combine to influence the powder dissolution behavior, resulting in little variability.
(11)$$∆S=(S_{\mathrm{CSDs}}-S_{\mathrm{drug}}{)/S}_{\mathrm{drug}}$$
where ∆S is the cumulative percentage dissolution rate of the CSD system relative to the drug, S_drug_ is the cumulative percentage dissolution rate of the drug after 4 h, and S_CSDs_ is the cumulative percentage dissolution rate of the CSD system with 30% drug loading after 4 h.

The relationship between interaction and crystal crystallite size is depicted in Figure 9. When a drug dissolves in a solvent to form a homogeneous solution, the crystallization process involves initial nucleation followed by crystal growth, which occurs after solvent evaporation using spray-drying techniques. However, a non-uniform state can emerge after solvent evaporation when the drug and polymer dissolve together and interact. In cases where the interaction between the drug and polymer is weak, the drug exhibits a crystallization trend similar to that of a pure drug. Conversely, when the interaction between the drug and polymer is strong, the polymer inhibits the drug’s crystal growth through a series of interaction mechanisms, resulting in relatively slow drug crystal growth and a reduction in crystal crystallite size.

## 4. Conclusions

In this study, we used various flavonoid analogs with different hydroxyl group numbers as model drugs and P188 as the carrier to explore the intrinsic relationship between CSDs’ intermolecular interactions, crystallite size, and dissolution behavior.

Firstly, we observed that as the number of hydroxyl groups increased, both solubility and interaction between flavonoid drugs and P188 also increased. We evaluated the relationship between the number of hydroxyl groups and intermolecular interactions, and we found that the number of hydroxyl groups was directly proportional to the strength of the interactions.

Secondly, we determined that the crystallization growth rate, crystalline domain size, and crystallite size of flavonoid drugs and P188 in CSDs was inversely related to the extent of their interaction. Moreover, the change in crystalline domain size (∆*τ*(%)) or crystallite size change rate (∆Dv(%)) was directly proportional to the extent of interaction; the larger the interaction, the greater the change rate.

Finally, the in vitro dissolution behavior of drugs in CSDs is proportional to the number of hydroxyl groups and the strength of the interactions. These findings provide theoretical support for future CSD preparation.

## Figures and Tables

**Figure 1 pharmaceutics-15-02493-f001:**
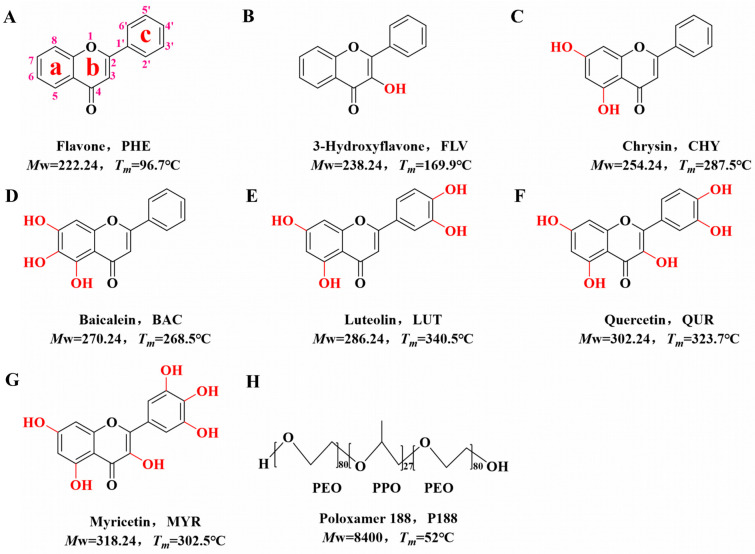
Chemical diagrams of model drugs and polymers. (**A**): PHE; (**B**): FLV; (**C**): CHY; (**D**): BAC; (**E**): LUT; (**F**): QUR; (**G**): MYR; (**H**): P188.

**Figure 2 pharmaceutics-15-02493-f002:**
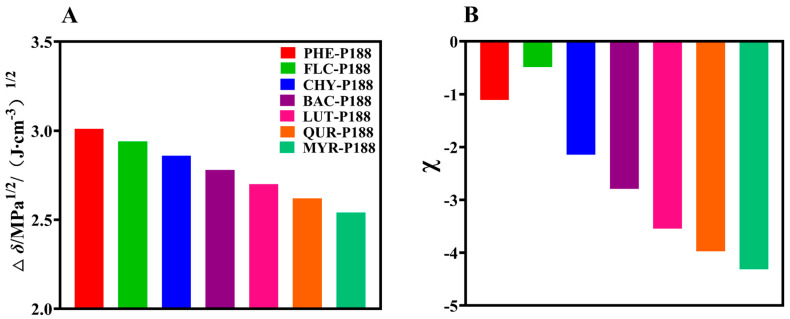
Determination of solubility parameters and interaction parameters. (**A**): Theoretical calculation of the solubility parameters; (**B**): Assessment of the Flory−Huggins interaction parameters.

**Figure 3 pharmaceutics-15-02493-f003:**
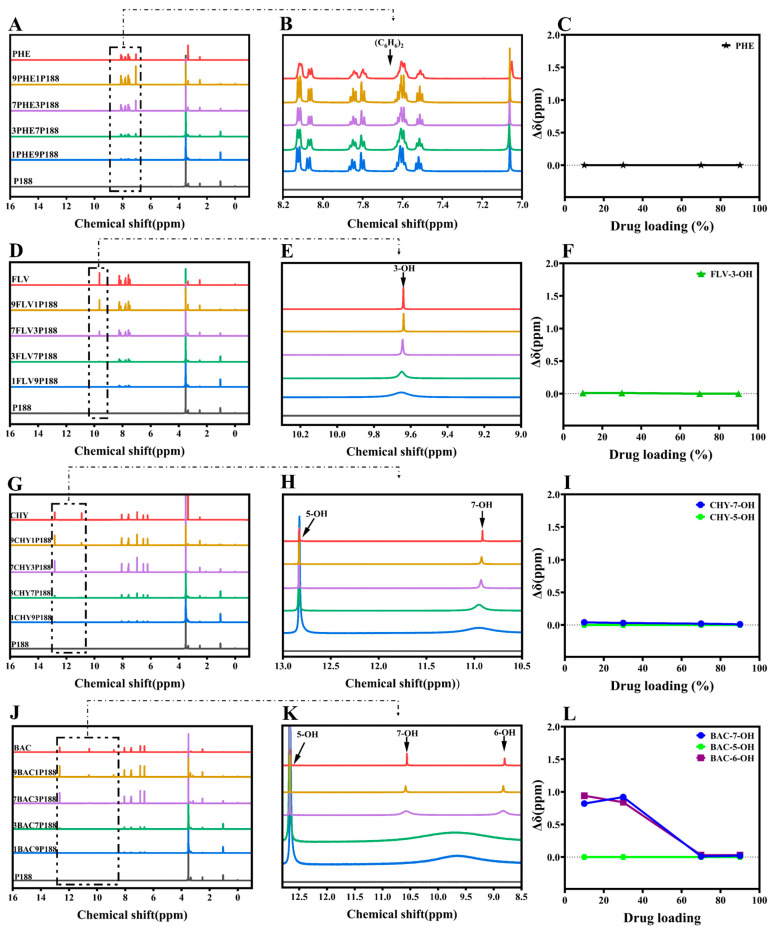
^1^H-NMR spectra of various flavonoid drugs mixed with P188. (**A**–**C**): Chemical shift variance in PHE; (**D**–**F**): Chemical shift variance in FLV; (**G**–**I**): Chemical shift variance in CHY; (**J**–**L**): Chemical shift variance in BAC; (**M**–**O**): Chemical shift variance in LUT; (**P**–**R**): Chemical shift variance in QUR; (**S**–**U**): Chemical shift variance in MYR.

**Figure 4 pharmaceutics-15-02493-f004:**
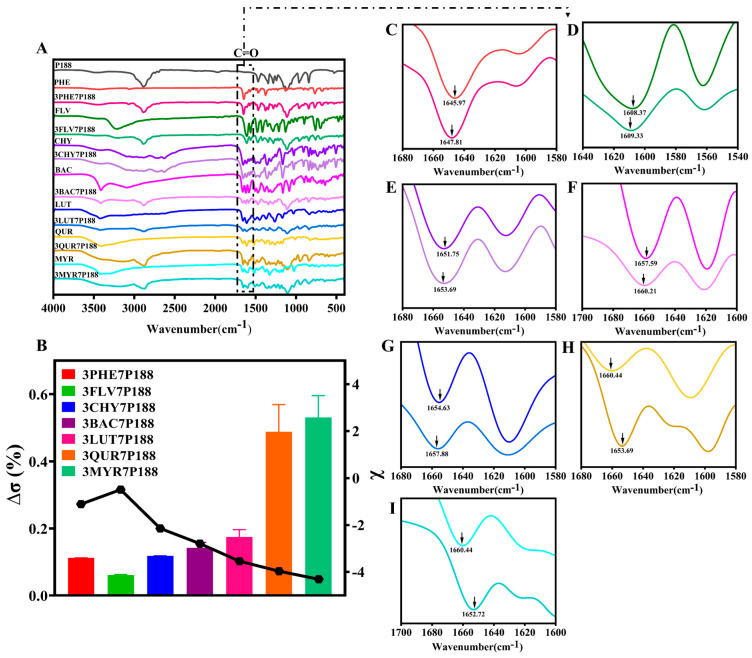
FT-IR spectra derived from flavonoids and the CSD system (30/70, *w*/*w*). (**A**): FT-IR analysis comparing flavonoids and CSD system (30/70, *w*/*w*); (**B**): Rate of change in the displacement of carbonyl peaks of flavonoids compared to the CSD system (discounted graphs representing drug–polymer interactions); (**C**): PHE; (**D**): FLV; (**E**): CHY; (**F**): BAC; (**G**): LUT; (**H**): QUR; (**I**): MYR.

**Figure 5 pharmaceutics-15-02493-f005:**
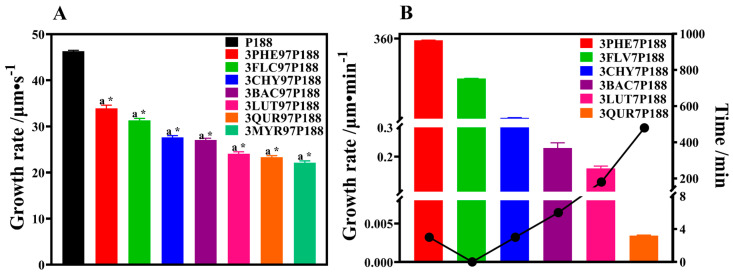
The growth rate of flavonoid drug crystals with P188. (The line graph in (**B**) indicates the time of appearance of the crystalline nuclei of the drug). (**A**): Crystallization growth rate of P188 with flavonoid drugs included at 3%; (**B**): Crystallization growth rate of flavonoid drugs included at 30%. a: *p* < 0.01 compared to P188; *: A chi−square trend test was used for linear associations between flavonoid drugs, *p* < 0.01.

**Figure 6 pharmaceutics-15-02493-f006:**
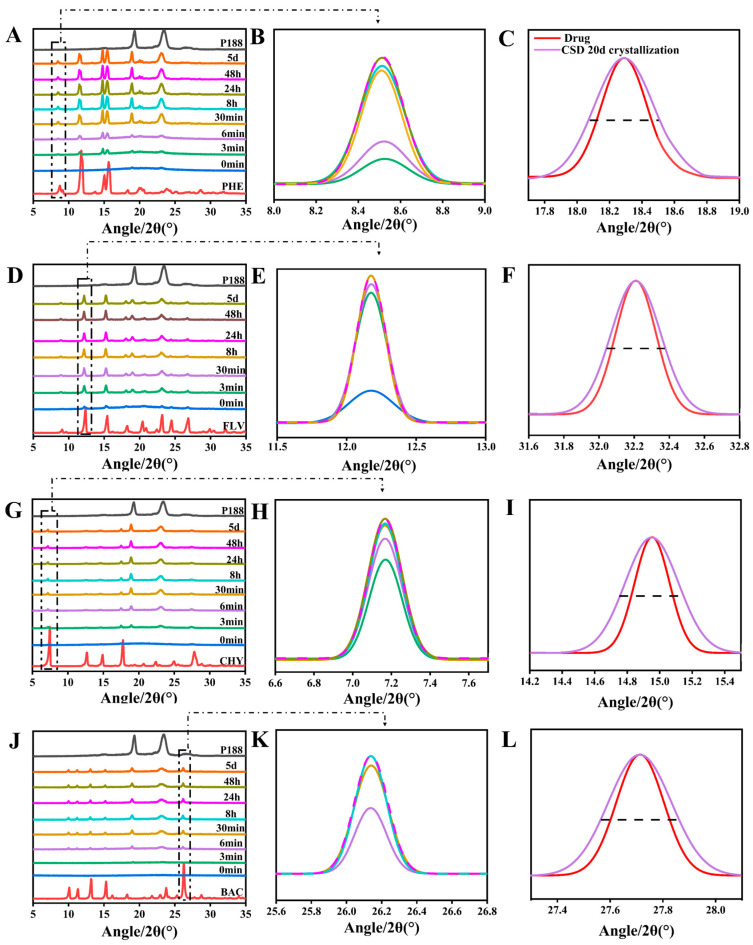
Crystallization kinetics and crystalline domain size variations in flavonoids within CSD systems loaded with 30% drug. (**A**/**B**,**D**/**E**,**G**/**H**,**J**/**K**,**M**/**N**,**P**/**Q**,**S**): Crystallization kinetics of PHE, FLV, CHY, BAC, LUT, QUR, MYR, respectively; (**C**,**F**,**I**,**L**,**O**,**R**,**T**): Crystalline domain size variations of PHE, FLV, CHY, BAC, LUT, QUR, MYR, respectively; (**U**): Compared with the CSD system, the domain size change rate of flavonoids (discounted graphs representing drug–polymer interactions).

**Figure 7 pharmaceutics-15-02493-f007:**
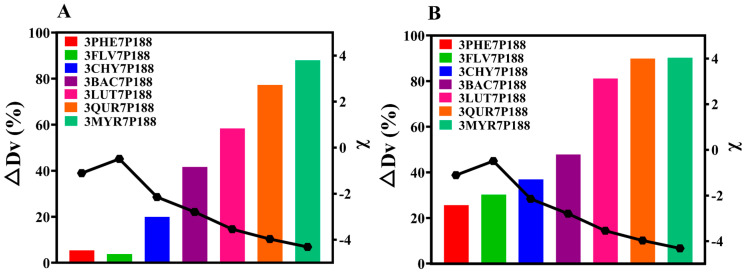
Particle-size alterations of flavonoids in the CSD system with 30% drug loading. (**A**): Particle-size variations in flavonoids in the CSD system with drug loading of 30% before washing; (**B**): Particle-size alterations of flavonoids in the CSD system with drug loading of 30% following washing with water. Discounted graphs representing drug–polymer interactions.

**Figure 8 pharmaceutics-15-02493-f008:**
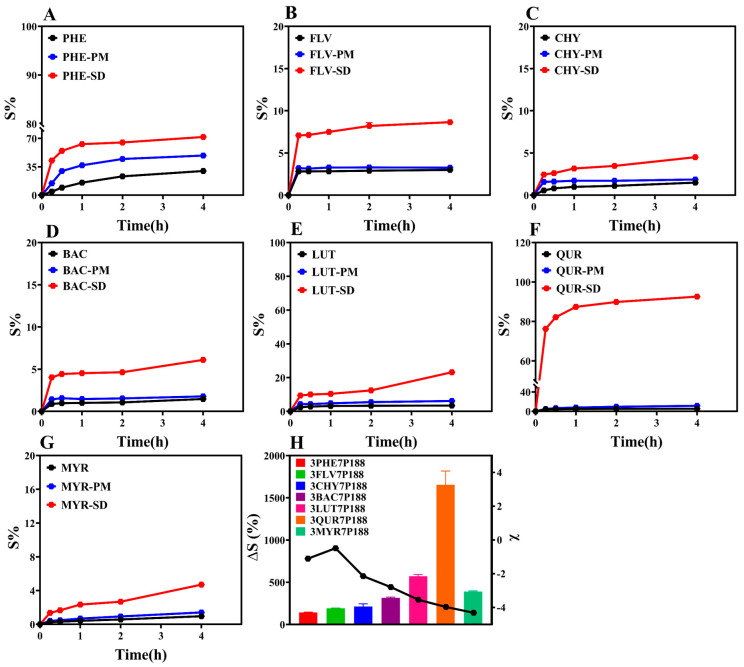
Cumulative dissolution rates of the drugs. (**A**): Cumulative dissolution rate of PHE; (**B**): Cumulative dissolution rate of FLV; (**C**): Cumulative dissolution rate of CHY; (**D**): Cumulative dissolution rate of BAC; (**E**): Cumulative dissolution rate of LUT; (**F**): Cumulative dissolution rate of QUR; (**G**): Cumulative dissolution rate of MYR; (**H**): Variation in powder dissolution in the CSD system with 30% drug loading after 4 h (discounted graphs representing drug–polymer interactions).

**Figure 9 pharmaceutics-15-02493-f009:**
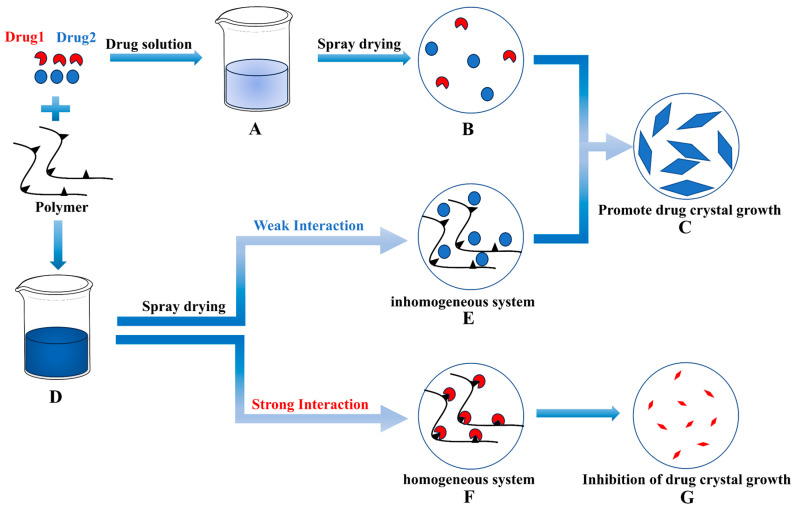
Correlation between interactions and crystallite size. (**A**,**D**): Homogenous solution; (**B**): Drug nucleation; (**C**): Drug growth; (**E**): Poorly interacting inhomogeneous systems; (**F**): Strongly interacting homogeneous systems; (**G**): Inhibition of drug growth.

## Data Availability

The datasets used or analyzed during the current study are available from the corresponding author upon reasonable request.

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
