# Peer review of "Influence of Intermolecular Interactions on Crystallite Size in Crystalline Solid Dispersions"

_pharmaceutics, 2023, doi:10.3390/pharmaceutics15102493_

Round 1

Reviewer 1 Report

The manuscript by Huang et al. studies the dependence between the number of hydroxyl groups in the flavonoid molecule and its interactions with triblock copolymer P188. Although the results seem convincing on a first glance, the detalied analysis of the manuscript raises questions to the presented conclusions. It is unclear how do the authors measure the 'degree of interaction' quantitatively, or would the trend persist when one of the components of the solid dispersion is changed. Therefore, I can advise the manuscript for major revision, with the list of issues given below:

Line 10 - The abbreiation CSD is unfortunate, since CSD is commonly referred to the Cambridge Structural Database, and a similar abbreviation is used in the anuscript for crystalline domain size.

Line 86 - 2-Phenylchromone can also be named flavone to highlight its role as the core compound for other flavonoids.

Figure 1 - While the choice of objects is based on increasing the number of hydroxyl groups, the positions of said groups differ among the selected compounds. Since the hydrogen bonding properties of OH groups at different positions are not identical, the substitution effect must be far from a simple linear dependence. The Reviewer wonders whether the described trand takes place would the authors choose different isomers for each number of OH groups.

Line 146 - Spectral assignments for flavonoid drugs and P188 were performed according to prior reports. The references for prior reports are needed

Figure 3 - It would be more representative to add a scheme displaying the interaction sites for target flavonoids derived from 1H NMR studies. A careful discussion of the obtained results is needed, since it can be seen from Figure 3 that e.g. the 7-OH position seems to be the primary binding site, followed by 4-OH position etc. A posible explanation of this order based on the molecular structures (bond polarity, electronic or steric effects) would make this observation a valuable addition to the manuscript.

Line 282 - It is not fully correct to compare the results of 1H NMR in solution with that of solid-state FTIR spectroscopy, since they refer to different states. A fraction of the hydrogen bonding sites in solution is occupied by solvent molecules, while for the solid, the competition is observed only between H-bonding groups of CSD components

Line 352 - The percentage change in crystallite domain size (Δτ (%))(Equation 9) was found to be directly proportional to the interactions (Figure 6U). - It is not clear what quantity is used as a measure of the interactions to plot correlations, since Figure 6U is a column chart. It is better to speck of the trend rather than correlation. The same for the direct proportionality with the degree of interaction (Line 413, Figure 8H)

Section 3.3 - There are more convenient ways exist to study drug-polymer interactions, as measuring the solubility of the drug in the solutions with inreasing concentrations of the polymer to determine the binding constant. The phase stability of the solid dispersion during the dissolution also needs to be investigated. The difference in solubilities of pure flavonoids also needs to be taken into account, as it obviously depends on the number of OH groups as well.

Minor language issues:

Line 143 - dissolved with deuterochloroform
Line 161 - Powders X-ray Diffraction
Line 162 - Powder X-ray diffraction (PXRD) orientations
Line 192 - the solution was removed,
Line 275 - Notably, 5-OH does not exhibit significant shifts in chemical shift
Line 369 - U: B : Compared with the CSD system, the domain size change rate of flavonoids.

Minor language issues:

Line 143 - dissolved with deuterochloroform
Line 161 - Powders X-ray Diffraction
Line 162 - Powder X-ray diffraction (PXRD) orientations
Line 192 - the solution was removed,
Line 275 - Notably, 5-OH does not exhibit significant shifts in chemical shift
Line 369 - U: B : Compared with the CSD system, the domain size change rate of flavonoids.

Author Response

Reviewer 1:

  1. Line 10 - The abbreiation CSD is unfortunate, since CSD is commonly referred to the Cambridge Structural Database, and a similar abbreviation is used in the anuscript for crystalline domain size.

Response: Thank you for your suggestions. First of all, I apologize for the reading inconvenience caused by my carelessness in writing. Secondly, we have abbreviated "Crystalline Solid Dispersions" to "CSDs", and the relevant changes have been highlighted in red in the new manuscript.

  1. Line 86 - 2-Phenylchromone can also be named flavone to highlight its role as the core compound for other flavonoids.

Response: I would appreciate your suggestions. In the new manuscript, we have changed "2-phenylchromanone" to " flavone", and the relevant changes have been highlighted in red in the new manuscript.(L76, L87, L97 Figure 1A)

  1. Figure 1 - While the choice of objects is based on increasing the number of hydroxyl groups, the positions of said groups differ among the selected compounds. Since the hydrogen bonding properties of OH groups at different positions are not identical, the substitution effect must be far from a simple linear dependence. The Reviewer wonders whether the described trand takes place would the authors choose different isomers for each number of OH groups.

Response: First of all, I would like to be grateful to you for your questions and suggestions, thanks to which I have identified insufficiencies in my current work and the suggestion is essential for my future research. Secondly, differences in hydroxyl positions may indeed affect drug polymer interactions; However, if this manuscript were to explore the topic of different hydroxyl positions while considering different numbers of hydroxyl groups, it would involve a number of experiments that would need to be completed, which could not be done within the editor's allotted time limit, and the main purpose of this study is to assess the correlation between the number of hydroxyl groups and intermolecular interactions. Your point about the effect of different hydroxyl positions on the differences in the formation of drug-polymer interactions in CSDs will be the next main focus of my work, which will be published in a subsequent article if there are further findings.

  1. Line 146 - Spectral assignments for flavonoid drugs and P188 were performed according to prior reports. The references for prior reports are needed

Response: First, we appreciate your questions and suggestions. Second, I have cited relevant references as you suggested, such as the following, and have supplemented them, and the relevant changes have been highlighted in red in the new manuscript. (L165)

  1. Baranović, G.; Šegota, S., Infrared spectroscopy of flavones and flavonols. Reexamination of the hydroxyl and carbonyl vibrations in relation to the interactions of flavonoids with membrane lipids. Spectrochimica Acta Part A: Molecular and Biomolecular Spectroscopy 2018, 192, 473-486.
  2. Wang, Y.; Li, L.; Cheng, G.; Li, L.; Liu, X.; Huang, Q., Preparation and recognition properties of molecularly imprinted nanofiber membrane of chrysin. Polymers 2022, 14, (12), 2398.
  3. Tong, M.; Wu, X.; Zhang, S.; Hua, D.; Li, S.; Yu, X.; Wang, J.; Zhang, Z., Application of TPGS as an efflux inhibitor and a plasticizer in baicalein solid dispersion. European Journal of Pharmaceutical Sciences 2022, 168, 106071.
  4. Alshehri, S.; Imam, S. S.; Altamimi, M. A.; Hussain, A.; Shakeel, F.; Elzayat, E.; Mohsin, K.; Ibrahim, M.; Alanazi, F., Enhanced dissolution of luteolin by solid dispersion prepared by different methods: physicochemical characterization and antioxidant activity. ACS omega 2020, 5, (12), 6461-6471.
  5. Van Hecke, E.; Benali, M., Solid dispersions of quercetin-PEG matrices: Miscibility prediction, preparation and characterization. Food Bioscience 2022, 49, 101868.
  6. Figure 3 - It would be more representative to add a scheme displaying the interaction sites for target flavonoids derived from 1H NMR studies. A careful discussion of the obtained results is needed, since it can be seen from Figure 3 that e.g. the 7-OH position seems to be the primary binding site, followed by 4-OH position etc. A posible explanation of this order based on the molecular structures (bond polarity, electronic or steric effects) would make this observation a valuable addition to the manuscript.

Response: First of all, thank you for your suggestion, which is of great importance for the improvement of this manuscript. Secondly, based on your suggestion, we have deeply discussed and explain the experimental results of 1H-NMR in the new manuscript, with the addition of " According to the experimental results of 1H-NMR, the chemical shifts of flavonoids prepared with P188 into the system of CSDs differs at different loading capacities. The results exhibited a more pronounced change in the chemical shifts of 7-OH, 6-OH and 4'-OH in the presence of multiple hydroxyl groups concomitantly, with 7-OH appearing to be the primary binding site for the drug-P188 interactions, followed by the 4'-OH position. In the structure of flavonoids with three six-membered rings, the a-ring is a conjugated system (Figure 1A), the introduction of hydroxyl group can form intramolecular hydrogen bonding, which increases the stability of the system, 7-OH is located in the 4-carbonyl position of the opposite position, susceptible to the conjugation of the carbonyl group induced by the p-π conjugation of the influence of the H+ leads to easier separation. The 5-OH is effected by the 4-carbonyl group to form intramolecular hydrogen bonds, where the H+ of the phenolic hydroxyl group is not easily ionized out. The stronger the hydrogen bonding acidity of a substance, the better its ability to act as a hydrogen bond donor and form intermolecular hydrogen bonds with other hydrogen bond acceptor substances, which can interact with polymers to a greater extent, where 7-OH is the main binding site for drug-P188 interaction. It is noticeable that there was no significant change in the chemical shift of 5-OH in the CSDs system of all flavonoids. Meanwhile, the a-ring is in the same plane as the b-ring, in which 3-OH can form an intramolecular hydrogen bond with the 4-carbonyl group. The c-ring is attached to the b-ring by a single bond at C-2 and can be rotated. When the hydroxyl group is present in the c-ring, which is affected by the electron-withdrawal effect of -OH. When the number of -OH gradually increases, the oxygen on the hydroxyl group has an unshared electron pair and can form a p-π conjugation with the benzene ring, which transfers the p-electrons on the oxygen to the benzene ring, resulting in easier ionization of H+ , with 4'-OH > 3'-OH ≈ 5'-OH in the c-ring." and the introduction of relevant references, which we expect to enrich the manuscript in terms of content, and the relevant changes have been highlighted in red in the new manuscript. (L292-L316)

  1. Liu, S.; Zhang, S.; Su, Y.; Liu, Q.; Liao, X., A Theoretical Study on Nuclear Magnetic Resonance Spectra of Three Flavonol Derivatives. CHINESE JOURNAL OF MAGNETIC RESONANCE 2007, 24, (2), 175.
  2. Biela, M.; Rimarčík, J.; Senajová, E.; Kleinová, A.; Klein, E., Antioxidant action of deprotonated flavonoids: Thermodynamics of sequential proton-loss electron-transfer. Phytochemistry 2020, 180, 112528.
  3. Janeiro, P.; Corduneanu, O.; Oliveira Brett, A. M., Chrysin and (±)‐taxifolin electrochemical oxidation mechanisms. Electroanalysis: An International Journal Devoted to Fundamental and Practical Aspects of Electroanalysis 2005, 17, (12), 1059-1064.
  4. Taft, R. W.; Kamlet, M. J., The solvatochromic comparison method. 2. The .alpha.-scale of solvent hydrogen-bond donor (HBD) acidities. Journal of the American Chemical Society 1976, 98, (10), 2886-2894.
  5. Seitsonen, A. P.; Idrissi, A.; Protti, S.; Mezzetti, A., Solvent effects on the vibrational spectrum of 3-hydroxyflavone. Journal of Molecular Liquids 2019, 275, 723-728.
  6. Ali, H. M.; Ali, I. H., Structure-antioxidant activity relationships, QSAR, DFT calculation, and mechanisms of flavones and flavonols. Medicinal Chemistry Research 2019, 28, 2262-2269.
  7. Barzegar, A., The role of intramolecular H-bonds predominant effects in myricetin higher antioxidant activity. Computational and Theoretical Chemistry 2017, 1115, 239-247.
  8. Line 282 - It is not fully correct to compare the results of 1H NMR in solution with that of solid-state FTIR spectroscopy, since they refer to different states. A fraction of the hydrogen bonding sites in solution is occupied by solvent molecules, while for the solid, the competition is observed only between H-bonding groups of CSD components.

Response: First of all, I appreciate you asking the question. Secondly, the previous studies confirmed that the drug-polymer interactions in the solid and liquid states do have some differences, so we examined the drug-polymer interactions in CSDs when they existed in the crystalline form on the basis of confirming the liquid interactions using 1H-NMR, to explain the drug-polymer interactions at a different levels and most of the conclusions of the study were reached from the interactions when they existed in crystalline form, and the relevant changes have been highlighted in red in the new manuscript. (L321-L324)

  1. Line 352 - The percentage change in crystallite domain size (Δτ (%))(Equation 9) was found to be directly proportional to the interactions (Figure 6U). - It is not clear what quantity is used as a measure of the interactions to plot correlations, since Figure 6U is a column chart. It is better to speck of the trend rather than correlation. The same for the direct proportionality with the degree of interaction (Line 413, Figure 8H).

Response: First of all, we would like to thank you for your suggestion, which is of great importance for the improvement of this manuscript. Secondly, we have modified the relevant graphs, as you suggested, by making the graphs as double Y-axis graphs, which directly correlate the rate of change of the crystalline sizes of the resulting CSDs with the interactions, and the relevant changes have been highlighted in red in the new manuscript. (L352 Figure 4B, L413 Figure 6H, L443 Figure 7, L479 Figure 8H)

  1. Section 3.3 - There are more convenient ways exist to study drug-polymer interactions, as measuring the solubility of the drug in the solutions with inreasing concentrations of the polymer to determine the binding constant. The phase stability of the solid dispersion during the dissolution also needs to be investigated. The difference in solubilities of pure flavonoids also needs to be taken into account, as it obviously depends on the number of OH groups as well.

Response: Regarding the issue of "using binding constants to determine interactions": Firstly, we are grateful for your suggestion, which provides us with a new idea for the future determination of interactions. Secondly, the article mainly focuses on drug-polymer interactions under solid state conditions; For the investigation of liquid drug-polymer interactions we have used 1H-NMR to explore the methodology; Furthermore, 1H-NMR is a more accurately and authoritative method to research liquid-state interactions and has been confirmed in several literatures. Therefore, in future studies, in which we will use the methodology you have proposed, in order to make our data more perfect.

Regarding the question of "the phase stability of the solid dispersion during the dissolution ": Firstly, I would like to thank you for your suggestion, which is of great importance to my future experimental design, and will be taken into account in our subsequent experimental design. Secondly, the focus of this article is on the effect of drug-polymer interactions on drug crystalline size, and the dissolution behavior is only to demonstrate that the strength of the interactions can lead to differences in the change in particle size, which in turn has an effect on the dissolution behavior. Finally, dissolution behavior was studied to demonstrate that designing insoluble drugs into CSDs is solubilizing. Therefore, the study of phase stability in dissolution behavior is not the main focus of this article and has not been investigated in detail.

Regarding the question of " The difference in solubilities of pure flavonoids": Firstly, thank you for your suggestions. Secondly, the physicochemical properties of each drug are different, and in order to better compare the data error caused by the change in solubility of the drug relative to CSDs (absolute value) when the drug is prepared into CSDs, we mainly interpret the change in the rate of change in the ratio of the difference of the drug relative to CSDs to the ratio of pure flavonoids (relative value) to the change in the CSDs.

Minor language issues:

  1. Line 143 - dissolved with deuterochloroform

Response: To begin with, thank you for your suggestion, and I apologize for the reading inconvenience caused by my carelessness in writing. Secondly, we have changed "deuterochloroform" to "dimethyl sulfoxide-d6+TMS(0.03)", and the relevant changes have been highlighted in red in the new manuscript. (L150, L152)

  1. Line 161 - Powders X-ray Diffraction

Line 162 - Powder X-ray diffraction (PXRD) orientations

Response: First of all, I would like to thank the reviewers for their careful observation, and I apologize for the reading inconvenience caused by my careless writing. Second, we have changed "Powder X-ray diffraction (PXRD) orientations " to " Powders X-ray diffraction (PXRD) patterns", and the relevant changes have been highlighted in red in the new manuscript. (L173)

  1. Line 192 - the solution was removed,

Response: Firstly, I would like to thank the reviewers for their criticisms and corrections, I deeply apologize for the reading inconvenience caused by my carelessness in writing. Secondly, we have revised "the solution was removed" to "0.3 mL of the sample was taken, and a blank dissolution medium of the same temperature and volume was added", and the relevant changes have been highlighted in red in the new manuscript. (L207-L208)

  1. Line 275 - Notably, 5-OH does not exhibit significant shifts in chemical shift

Response: First of all, I would like to thank the reviewers for their careful observation, and I apologize for the reading inconvenience caused by my careless writing. Secondly, we have revised "Notably, 5-OH does not exhibit significant shifts in chemical shift" to " The 5-OH is effected by the 4-carbonyl group to form intramolecular hydrogen bonds, where the H+ of the phenolic hydroxyl group is not easily ionized out", and the relevant changes have been highlighted in red in the new manuscript. (L302-L303)

  1. Line 369 - U: B : Compared with the CSD system, the domain size change rate of flavonoids.

Response: Firstly, I would like to thank the reviewers for their criticisms and corrections, I deeply apologize for the reading inconvenience caused by my carelessness in writing. Secondly, we have deleted "B :", and the relevant changes have been highlighted in red in the new manuscript. (L418)

Reviewer 2 Report

This paper is sufficiently impressive. It was presented clearly. However, there are still some issues that require clarification and improvement. The details of the correction are set forth below.
1. 
Typographical errors must be corrected.

2. In the current version, the abstract is too long, the conclusion as well.

3.The paper is well arranged. The authors should be concise in the introduction part. Some previously studied  should be added.

4. The quality of some figures should improved.

The English language should be improved.

Author Response

Reviewer 2 :

  1. Typographical errors must be corrected.

Response: We appreciate your suggestions and we tried our best to improve the manuscript and made some corrections to it. These modifications do not affect the content or framework of the manuscript. Instead of listing these changes here, we have highlighted them in red in the revised manuscript. We express our sincere gratitude to the editors/reviewers for their dedicated work and expect that the revisions will be recognized. (For example, in L150, L173, L418)

  1. In the current version, the abstract is too long, the conclusion as well.

Response: Firstly, thank you for making the suggestion. Secondly, we have condensed and improved the abstract and conclusions, and the relevant changes have been highlighted in red in the new manuscript. (L10-L25, L501-515)

  1. The paper is well arranged. The authors should be concise in the introduction part. Some previously studied should be added.

Response: Firstly, I would like to thank you for your suggestions, which are very important for the improvement of this article. Secondly, in accordance with your suggestion, we have summarized and supplemented the content of the previously studied in the "Introduction", and introduced relevant references as support, so as to enrich the content of the preface part of the manuscript, and the relevant changes have been highlighted in red in the new manuscript. (L43-L48, L59-L62, L75-L76)

  1. Hu, C.; Liu, Z.; Liu, C.; Zhang, Y.; Fan, H.; Qian, F., Improvement of Antialveolar Echinococcosis Efficacy of Albendazole by a Novel Nanocrystalline Formulation with Enhanced Oral Bioavailability. ACS Infect Dis 2020, 6, (5), 802-810.
  2. Roduner, E., Size matters: why nanomaterials are different. Chem Soc Rev 2006, 35, (7), 583-92.
  3. Mihranyan, A.; Strømme, M., Solubility of fractal nanoparticles. Surface Science 2007, 601, (2), 315-319.
  4. Galli, C., Experimental determination of the diffusion boundary layer width of micron and submicron particles. International journal of pharmaceutics 2006, 313, (1-2), 114-122.
  5. Chen, Z.; Liu, Z.; Qian, F., Crystallization of bifonazole and acetaminophen within the matrix of semicrystalline, PEO–PPO–PEO triblock copolymers. Molecular Pharmaceutics 2015, 12, (2), 590-599.
  6. Pool, H.; Quintanar, D.; de Dios Figueroa, J.; Mano, C. M.; Bechara, J. E. H.; Godínez, L. A.; Mendoza, S., Antioxidant effects of quercetin and catechin encapsulated into PLGA nanoparticles. Journal of nanomaterials 2012, 2012, 86-86.
  7. Zheng, Y.-Z.; Zhou, Y.; Liang, Q.; Chen, D.-F.; Guo, R., A theoretical study on the hydrogen-bonding interactions between flavonoids and ethanol/water. Journal of molecular modeling 2016, 22, 1-10.
  8. The quality of some figures should improved.

Response: Firstly, thank you for making the suggestion. Secondly, as you suggested, the figures in the article have been modified to improve the quality, and the relevant changes have been highlighted in red in the new manuscript. (L28, L98 Figure 1, L240 Figure 2, L279 Figure 3, L353 Figure 4, L383 Figure 5, L414 Figure 6, L444 Figure 7, L479 Figure 8)

Reviewer 3 Report

Manuscript ID: pharmaceutics-2637667

Title: Influence of Intermolecular Interactions on Crystallite Size in Crystalline Solid Dispersions 

Authors: Hua Huang, Yong Zhang, Yao Liu, Yufei Guo, Chunhui Hu *

The purpose of this work was to study crystallization processes in composite molecular systems. Mixtures of 2-phenylchromone derivatives and the polymer Poloxamer 188 served as model systems. The results of the study showed that an increase in the number of active centers through which the components of the mixture can interact with each other leads to a decrease in the rate of crystallization and a decrease in the size of crystalline domains. The purpose of the study is justified. The conclusions drawn by the authors are confirmed by experimental results.

Changes required.

1.     Line 145. “The deuterochloroform solvent signal was utilized as a reference (DMSO-d6).” The meaning of the sentence is not clear.

2.     Report the molecular and polymer concentrations for each NMR sample. Without this information, the comparison of the obtained NMR spectra is meaningless.

3.     Line 282. The fact that the vibration frequency of C=O is different in the pure substances and in their mixtures with the polymer indicates that complete crystallization of the substances has not yet occurred in these mixtures. Thus, the magnitude of the observed differences depends not so much on the strength of interactions as on the degree of crystallization at the time of measurement.

4.     Line 345-347. “The crystallization times for FLV, PHE, CHY, BAC, LUT, and QUR were 0 minutes, 3 minutes, 3 minutes, 3 hours, and 8 hours, and MYR showed no crystallization within 5 days, indicating that more extensive interactions result in slower drug crystallization.” The indicated times correspond to periods after which the appearance of crystalline nuclei in the mixture could be detected experimentally. It is necessary to clarify this fact explicitly. In addition, it must be explained to the reader that in such systems different dynamic processes compete and the more different interactions are possible, the slower equilibrium is achieved. (see: DOI: 10.1002/mrc.932, 10.1039/B617744A, or any other publication on this topic.)

5.     Line 403. Myricetin exhibits the lowest degree of crystallization. Why is the degree of its dissolution so low?

6.     Line 448-450. “Firstly, based on Hansen solubility parameters and Flory-Huggins interaction parameters, we observed that as the number of hydroxyl groups increased, the solubility and interaction between flavonoid drugs and P188 also increased.” Do you mean the solubility of the drugs or drug/P188 complexes?

Author Response

Reviewer 3 :

  1. Line 145. “The deuterochloroform solvent signal was utilized as a reference (DMSO-d6).” The meaning of the sentence is not clear.

Response: To begin with, thank you for your suggestion, and I apologize for the reading inconvenience caused by my carelessness in writing. Secondly, we have changed "deuterochloroform" to "dimethyl sulfoxide-d6+TMS(0.03)", and the relevant changes have been highlighted in red in the new manuscript. (L150, L152)

  1. Report the molecular and polymer concentrations for each NMR sample. Without this information, the comparison of the obtained NMR spectra is meaningless.

Response: Firstly, thank you for your suggestion and question, which is very important for the improvement of this article. Secondly, I apologize for the reading inconvenience caused by my carelessness in writing. Finally, I have added to the experimental design of 1H-NMR: " In the experimental design, the mass of the pure drug was consistently kept at 10 mg throughout the experiment in order to avoid multiple variables in the measurements, which would ultimately lead to errors caused by other factors", and the relevant changes have been highlighted in red in the new manuscript. (L154-157)

  1. Line 282. The fact that the vibration frequency of C=O is different in the pure substances and in their mixtures with the polymer indicates that complete crystallization of the substances has not yet occurred in these mixtures. Thus, the magnitude of the observed differences depends not so much on the strength of interactions as on the degree of crystallization at the time of measurement.

Response: Firstly, thank you for your suggestion, and I apologize for the reading inconvenience caused by my carelessness in writing. Secondly, I have revised "the strength of interactions" to "the degree of crystallization" according to your suggestion, and added the relationship between interactions and degree of crystallization, as follows: "the strength of interactions" to "the degree of crystallization". I have also added the relationship between interactions and the degree of crystallization, which reads: " Differences in the degree of crystallization are also partially due to differences in the interactions. It was verified from the aspect that the difference in interactions affects their crystallization behavior." Relevant expressions in the article were modified and added, and the relevant changes have been highlighted in red in the new manuscript. (L325, L326, L327-L329)

  1. Chen, Z.; Liu, Z.; Qian, F., Crystallization of bifonazole and acetaminophen within the matrix of semicrystalline, PEO–PPO–PEO triblock copolymers. Molecular Pharmaceutics 2015, 12, (2), 590-599.

  1. Line 345-347. “The crystallization times for FLV, PHE, CHY, BAC, LUT, and QUR were 0 minutes, 3 minutes, 3 minutes, 3 hours, and 8 hours, and MYR showed no crystallization within 5 days, indicating that more extensive interactions result in slower drug crystallization.” The indicated times correspond to periods after which the appearance of crystalline nuclei in the mixture could be detected experimentally. It is necessary to clarify this fact explicitly. In addition, it must be explained to the reader that in such systems different dynamic processes compete and the more different interactions are possible, the slower equilibrium is achieved. (see: DOI: 10.1002/mrc.932, 10.1039/B617744A, or any other publication on this topic.)

Response: Firstly, thank you for your suggestion and question, and I apologize for the reading inconvenience caused by my carelessness in writing. Secondly, I have revised "crystallization times" to "the appearance of crystalline nuclei in the mixture" as suggested by you; and added interactions and interactions in the mixture. "and added the link between interactions and equilibrium as " It must be explained that in such systems different dynamic processes compete and the more different interactions are possible, the slower equilibrium is achieved ", and supplemented with relevant reference, and the relevant changes have been highlighted in red in the new manuscript. (L389-L390, L392, L394, L395-397)

  1. Gedat, E.; Schreiber, A.; Findenegg, G. H.; Shenderovich, I.; Limbach, H. H.; Buntkowsky, G., Stray field gradient NMR reveals effects of hydrogen bonding on diffusion coefficients of pyridine in mesoporous silica. Magnetic Resonance in Chemistry 2010, 39, (S1). DOI:10.1002/mrc.932.
  2. Line 403. Myricetin exhibits the lowest degree of crystallization. Why is the degree of its dissolution so low?

Response: First, thank you for your question. Secondly, the lower degree of dissolution of MYR may be due to the following reasons: 1) Re-search suggests that its in vitro dissolution is related to drug loading. While 30% of drug loading was selected for the study in order to standardize the drug loading in this study, it may not be the optimal drug loading in the dissolution of MYR under this condition, which does not reflect the optimal dissolution behavior; 2) The powder dissolution behavior of a drug is affected by a number of factors, such as the particle size of the drug, the nature of the drug, with dissolution conditions, and etc. Whereas this study mainly illustrates the variability in terms of variation in drug particle size, hence there may be other factors that combine to influence the powder dis-solution behavior, resulting in little variability. The next study will be conducted later to prove this conclusion, and the relevant changes have been highlighted in red in the new manuscript. (L466-L475)

  1. Line 448-450. “Firstly, based on Hansen solubility parameters and Flory-Huggins interaction parameters, we observed that as the number of hydroxyl groups increased, the solubility and interaction between flavonoid drugs and P188 also increased.” Do you mean the solubility of the drugs or drug/P188 complexes?

Response: Firstly, thank you for your question, and I apologize for any misinterpretation of your reading due to my carelessness in writing. Secondly, we are referring to the "solubility of flavonoid drug-P188 complexes", and the relevant changes have been highlighted in red in the new manuscript. (L505)

Round 2

Reviewer 1 Report

The authors have corrected most of the issues, and some discussion was added, which improved the scientific quality of the manuscript. However, the text is still raw and requires a thorough and careful reading to remove the remaining errors, grammar mistakes and typos. An incomplete list of found issues is given below:

Line 25 - The dissolution enhancement rather than solubility

Line 88 - The correct compound name is 3-hydroxyflavone, not 3-hydroxyflavonol. Flavan-3-ol is a proper synonym (see Figure 1B)

Line 144 - Tm0 should stand instead of Tm. Please use lower index.

Line 152 - Please specify the abbreviation TMS(0.03)

Line 162 - In the future publications, please use higher IR spectral resolution

Line 195 - It would be nice to provide the equation for particle size from sample refraction

Line 322 - Please add references for previous studies

Line 394 - The nucleation times could be added as the second layer to Figure 5B

Figure 6 A-T - Please add references to panels and a brief discussion in the manuscript

Language issues:

Line 18 and so on - The correct version is powder (no plural) X-ray difraction
Line 46 - 'theoretical theories' is excessive
Line 91 - Poroxamer P188 is a typo
Line 390 - the verb is missing

Line 18 and so on - The correct version is powder (no plural) X-ray difraction
Line 46 - 'theoretical theories' is excessive
Line 91 - Poroxamer P188 is a typo
Line 390 - the verb is missing

Author Response

  1. Line 25 - The dissolution enhancement rather than solubility

Response: We sincerely thank the reviewer for careful reading. As suggested by the reviewer, we have corrected the " solubility " into " dissolution ", and the relevant changes have been highlighted in yellow in the new manuscript. (L25)

  1. Line 88 - The correct compound name is 3-hydroxyflavone, not 3-hydroxyflavonol. Flavan-3-ol is a proper synonym (see Figure 1B)

Response: Firstly, I would like to thank the reviewers for their criticisms and corrections, I deeply apologize for the reading inconvenience caused by my carelessness in writing. Secondly, we have revised " 3-hydroxyflavonol " to " 3-hydroxyflavone ", and the relevant changes have been highlighted in yellow in the new manuscript. (L88, L97 Figure 1B)

  1. Line 144 - Tm0 should stand instead of Tm. Please use lower index.

Response: Firstly, I would appreciate your suggestions. Secondly, we have changed " Tm0" to " Tm", and the relevant changes have been highlighted in yellow in the new manuscript. (L142-L144)

  1. Line 152 - Please specify the abbreviation TMS (0.03)

Response: Firstly, I would appreciate your suggestions. Secondly, we have added to the suggestions you have made by clarifying our definition of TMS, and the relevant changes have been highlighted in yellow in the new manuscript. (L151-L152)

  1. Line 162 - In the future publications, please use higher IR spectral resolution

Response: Firstly, I would like to be grateful to you for your suggestions, and the suggestion is essential for my future research. Secondly, in future studies we will use higher resolution to improve the quality of our data which will make our data more accurate and enrichment.

  1. Line 195 - It would be nice to provide the equation for particle size from sample refraction

Response: First, we appreciate your suggestion. Secondly, by reviewing the literature, we could only find the formula of the light scattering principle (Mie principle) to indirectly calculate the drug particle size and the drug distribution, which due to the time constraints, the researchers are not convinced that this is a formula that can accurately be reflected in the relationship between the particle size and the refractive index. Furthermore, the refractive index and size of the particles were not discussed in this study, so if this formula was included in the manuscript it would affect the coherence and readability of the manuscript, so we did not add this formula to the manuscript. The specific laser particle sizer principle and the detailed expression of the Mie equation are as follows:

in which λ denotes the wavelength of light in the medium, θ denotes the scattering angle, I0 denotes the incident light intensity, φ denotes the angle between the direction of polarization of the incident light and the scattering surface, r denotes the distance from the observation to the scattering particles, and s1(θ) denotes the amplitude of the scattered light perpendicular to the scattering surface, and s2(θ) denotes the amplitude of the scattered light parallel to the scattering surface.

  1. Line 322 - Please add references for previous studies

Response: Firstly, thank you for your suggestions, which are of great importance to the improvement of this article. Secondly, in accordance with your suggestion, reference has been introduced into the relevant part of the manuscript as a support to enrich the content of the manuscript part, and the relevant changes have been highlighted in yellow in the new manuscript. (L324)

  1. Chen, Y.; Liu, C.; Chen, Z.; Su, C.; Hageman, M.; Hussain, M.; Haskell, R.; Stefanski, K.; Qian, F., Drug–polymer–water interaction and its implication for the dissolution performance of amorphous solid dispersions. Molecular pharmaceutics 2015, 12, (2), 576-589.
  2. Line 394 - The nucleation times could be added as the second layer to Figure 5B

Response: Firstly, thank you for your suggestions, which are of great importance to the improvement of this article. Secondly, we have revised the diagram in the relevant part of Figure 5B in the manuscript, and the relevant changes have been highlighted in yellow in the new manuscript. (L383-L385)

  1. Figure 6 A-T - Please add references to panels and a brief discussion in the manuscript

Response: Firstly, we appreciate your suggestions, which are of great significance to the improvement of this article. Secondly, according to your suggestions, the study has been summarized and supplemented in the relevant part, and relevant literature has been introduced as a support to make the contents of the preface part of the manuscript more substantial, and the relevant changes have been highlighted in yellow in the new manuscript. (L391-393, L402-405, L416-417, L421-425)

  1. Chen, Z.; Liu, Z.; Qian, F., Crystallization of bifonazole and acetaminophen within the matrix of semicrystalline, PEO–PPO–PEO triblock copolymers. Molecular Pharmaceutics 2015, 12, (2), 590-599.
  2. Taylor, L. S.; Zografi, G., Spectroscopic characterization of interactions between PVP and indomethacin in amorphous molecular dispersions. Pharmaceutical research 1997, 14, 1691-1698.

Comments on the Quality of English Language

Language issues:

  1. Line 18 and so on - The correct version is powder (no plural) X-ray diffraction

Response: Firstly, I would like to thank the reviewers for their criticisms and corrections, I deeply apologize for the reading inconvenience caused by my carelessness in writing. Secondly, we have changed " powders X-ray diffraction " to " powder X-ray diffraction ", and the relevant changes have been highlighted in yellow in the new manuscript. (L18, L19, L173, L174)

  1. Line 46 - 'theoretical theories' is excessive

Response: Firstly, I would like to thank the reviewers for their careful observation, and I apologize for the reading inconvenience caused by my careless writing. Secondly, we have deleted " theoretical ", and the relevant changes have been highlighted in yellow in the new manuscript. (L46)

  1. Line 91 - Poroxamer P188 is a typo

Response: Firstly, I would like to thank the reviewers for their careful observation, and I apologize for the reading inconvenience caused by my careless writing. Secondly, we have deleted " P ", and the relevant changes have been highlighted in yellow in the new manuscript. (L90)

  1. Line 390 - the verb is missing

Response: Firstly, I would like to thank the reviewers for their criticisms and corrections, I deeply apologize for the reading inconvenience caused by my carelessness in writing. Secondly, we have rewritten the sentence and added related verbs, and the relevant changes have been highlighted in yellow in the new manuscript. (L3880-L389)

Reviewer 3 Report

References must be in the required format: Journal, year, volume, page. 

Especially see Ref.  2, 24, 27, 66

Author Response

References must be in the required format: Journal, year, volume, page. 

  1. Especially see Ref. 2, 24, 27, 66

Response: Firstly, I would like to thank the reviewers for their criticisms and corrections, I deeply apologize for the reading inconvenience caused by my carelessness in writing. Secondly, we have corrected the citation format of the references in the manuscript, and the relevant changes have been highlighted in yellow in the new manuscript. (L545, L594, L603, L702)
